# Starch Edible Films/Coatings Added with Carvacrol and Thymol: In Vitro and In Vivo Evaluation against *Colletotrichum gloeosporioides*

**DOI:** 10.3390/foods10010175

**Published:** 2021-01-16

**Authors:** Carlos Enrique Ochoa-Velasco, Julio César Pérez-Pérez, José Mauricio Varillas-Torres, Addí Rhode Navarro-Cruz, Paola Hernández-Carranza, Ricardo Munguía-Pérez, Teresa Soledad Cid-Pérez, Raúl Avila-Sosa

**Affiliations:** 1Facultad de Ciencias Químicas, Benemérita Universidad Autónoma de Puebla, Puebla 72420, Mexico; carlos.ochoa@correo.buap.mx (C.E.O.-V.); juliocesar.perez@correo.buap.mx (J.C.P.-P.); mauricio.varillas@correo.buap.mx (J.M.V.-T.); addi.navarro@correo.buap.mx (A.R.N.-C.); paola.hernandez@correo.buap.mx (P.H.-C.); teresa.cid@correo.buap.mx (T.S.C.-P.); 2Centro de Investigaciones en Ciencias Microbiológicas, Laboratorio de Micología, Instituto de Ciencias, Benemérita Universidad Autónoma de Puebla, Puebla 72420, Mexico; ricardo.munguia@correo.buap.mx

**Keywords:** anthracnose, synergism, natural antimicrobials, active packaging

## Abstract

The aim of this work was to evaluate the in vitro and in vivo effectiveness of thymol and carvacrol added to edible starch films and coatings against *Colletotrichum gloeosporioides*. In vitro evaluation consisted of determining minimal inhibitory concentration (MIC) of carvacrol and thymol was determined at different pH values against *Colletotrichum gloeosporioides*. With MIC values, binary mixtures were developed. From these results, two coatings formulations were in vivo evaluated on mango and papaya. Physicochemical analysis, color change, fruit lesions and *C. gloeosporioides* growth were determined during storage. In vitro assay indicated that the MIC value of carvacrol and thymol against *C. gloeosporioides* was 1500 mg/L at pH 5. An additive effect was determined with 750/750 and 1125/375 mg/L mixtures of carvacrol and thymol, respectively. Coated fruits with selected mixtures of carvacrol and thymol presented a delay in firmness, maturity index and color change. Moreover, a fungistatic effect was observed due to a reduction of lesions in coated fruits. These results were corroborated by the increase in the lag phase value and the reduction of the growth rate. Carvacrol and thymol incorporated into edible films and coatings are able to reduce the incidence of anthracnose symptoms on mango and papaya.

## 1. Introduction

Fruit global production has been increasing as a result of world population demand [1]. However, the majority of fresh fruits are susceptible to infection by some pathogenic fungi during the postharvest period. Contamination by some filamentous fungi represents the primary cause of rapid spoilage of fresh fruits, which affects their quality and decreases shelf life [2,3]. Fresh product losses are estimated to reach 20–50% in developing countries and up to 25% in developed nations [4]. One of the most widespread fungal diseases is anthracnose, a cosmopolitan infection for many tropical and subtropical fruits. It is widely recorded crop loss in preharvest and postharvest caused by *Colletotrichum gloeosporioides* [5]. Currently, this genus was recognized as the most prevalent fungal pathogens causing diseases in diverse tropical and subtropical fruits [5,6]. Although chemical fungicides are frequently used for controlling fungal pathogen affectation, their application is currently restricted. However, their use is increasingly restricted due to public concerns about possible toxicological risks for people. The environment is associated with excessive chemical residues and trends of resistance against fungicides, thereby shortening the lifespan of protective products [4,7,8]. Consequently, alternative strategies for reducing the occurrence of postharvest disease in the fruit have been demanded. It is essential that technologies not only ensure food quality or increase shelf life, but in addition, receive a reduced environmental impact [9,10].

Essential oils (EO) obtained from plants or spices are excellent sources of biologically active compounds, like terpenoids and phenolic acids; among them, carvacrol and thymol are present in oregano EO. These compounds are recognized as antimicrobials [11]. This characteristic has already been studied and reported against several bacterial and fungal strains [12,13,14,15]. Thymol is structurally similar to carvacrol, though it has the hydroxyl group at a different location on the phenolic ring [16]. In general, antimicrobials cause structural and functional damages to microorganisms by interacting with the microbial cell membrane dissipate H^+^ and K^+^ ion gradients, allowing leakage of vital cellular constituents. Moreover, they have been registered by the European Commission for use as a food additive in view of its non-toxic effect on the mammalian system and preservative potential [17]. An inconvenience of EO direct application on food surfaces by dipping, powdering or spraying; is that antimicrobial compounds can be neutralized, evaporated or rapidly diffused from the surface into the product [18,19]. For instance, thymol showed antifungal activity in the vapor phase [20], with synergistic effects with other terpenes against different fungal genera [21]. Thus, an alternative to carry out this kind of additives is the use of edible films and coatings.

Edible films and coatings are thin layers of material (proteins, polysaccharides and/or lipids) that offers the potential to provide: a selective barrier to moisture, carbon dioxide and oxygen, improve mechanical and textural properties, prevent flavorlessness, and act as a carrier for different food additives to avoid pathogen infection [22,23]. Moreover, some researchers demonstrated the potential of edible antimicrobial coatings in fresh fruits to increase their shelf life [24]. A preponderant factor for developing this technology is due to its nontoxicity, biocompatibility, biodegradability and reproducibility properties [23,25,26]. The efficiency of fruit coating materials primarily depends on the coatings ingredients’ nature and their relative optimal concentration [27].

Starch is a polysaccharide consisting of mostly linear chains of amylose (ideal for film-forming) and highly branched chains of amylopectin and derived from diverse resources. Starch is cheap, commercially available, transparent, colorless and tasteless [25,28]. Moreover, the use of starch and its derivates for producing edible films or coatings is extensive. Generally, such coatings are less costly than alternative high tensile strength materials. Various compounds like glycerol improve starch-based coatings’ functional properties. It allows thymol and carvacrol dispersion increasing the particle distribution among the edible films [29,30].

The aim of this work was to evaluate in vitro, MIC values and binary mixtures effectiveness of thymol and carvacrol added to edible starch films at different pH and in vivo, effective binary mixtures effectiveness, the physicochemical parameter of edible coatings against *C. gloeosporioides*.

## 2. Materials and Methods

### 2.1. Plant Material

Mango (*Mangifera indica* var. Ataulfo) and papaya (*Carica papaya* L. var. Maradol) were purchased from a local market in Puebla, Mexico. Fruits were chosen free from physical and microbiological appearance damage and with a homogenous mature stage (according to the color). Fruits were disinfected with a sodium hypochlorite solution (150 g/L) for 1 min and immediately dried with absorbent paper.

### 2.2. Reagents and Culture Media

Reagents and culture media used in this study were purchased from Sigma-Aldrich, Inc. (Milwaukee, WI, USA) and Merk (Mexico City, Mexico).

### 2.3. In Vitro Evaluation of Thymol and Carvacrol Added to Starch Edible Films against C. gloeosporioides

*C. gloeosporioides* was isolated from infected papaya fruits (*Carica papaya* L.). Fruits with showing lesions and characteristic symptoms of fungal infection were collected at a local market. Fruits were washed with tap water, and minor portions (0.5 cm^2^) of contaminated epidermis were cut off and then disinfected with a 1% solution of sodium hypochlorite for 1 min. After disinfection, tissue portions were washed three times with sterile distilled water to eliminate chloride residues. Each portion cut was placed in Petri dishes (100 × 15 mm) containing potato-dextrose agar plates (PDA). Dishes were stored in a dark environment at 28 °C for 5–8 days until fungal growth was observed. Fungus structure (conidia and mycelia) was observed with an optical microscope (Zeiss Primo Star, Göttingen, Germany), and identification was according to the published taxonomic key [31]. Once *C. gloeosporioides* was isolated, a suspension containing 2500 spores/mL (using a hemocytometer) was prepared from the fungal growth to purify it and by means of a capillary tube. Spores were transferred to the center of another Petri dish containing PDA medium, which was incubated in the dark for 8 days at 28 °C [32]. To recover fungal spores, sterile physiological water was poured on the growth agar plate surface, followed by a gentle scraping using a sterile rake to remove the maximum quantity of spores. After this, spore suspensions were transferred into sterile tubes. The number of spores present in the suspension was determined using a hemocytometer and an optical microscope (Zeiss Primo Star, Göttingen, Germany) and expressed as the number of spores per milliliter (spores/mL) [33].

For film preparation and casting, one g of high amylose corn starch was mixed with 10 mL of previously sterilized 0.25 N sodium hydroxide and 10 mL of distilled water. Film-forming solutions (FFS) were maintained 60 min under stirring conditions. At that point, it was gelatinized in a shaker water bath at 78–80 °C for 10 min; when the solution reached 40 °C, glycerol (1.2% *v*/*v*) was added [34]. pH was adjusted to 4, 5, 6 or 7 with phosphoric acid (1 N) and mixed under aseptic conditions at 20,000 rpm (IKA high-performance disperser T18, Chicago, IL, USA) for 1 min at room temperature. Thymol or carvacrol was added to reach a final concentration of 0, 250, 500, 750, 1000, 1250, 1500, 1750, 2000, 2500, or 3000 (mg/L). Films were prepared with 7 mL of FFS per dish (60 mm inner diameter sterile Petri dishes) and dried under 0.35 kg/cm^2^ vacuum at 30 °C for 12 h. Films were cut in 5 mm discs under sterile conditions and kept in sealed Petri dishes at 4 °C until studies.

To evaluate the antifungal effect, spore suspension (30 μL) was inoculated onto agar surface; after 30 min, film discs were placed on solidified PDA agar plates and incubated at 25 °C. A growth control was prepared in parallel; radial growth was measured every 24 h for 7 d. Every test was performed in triplicate. Minimal inhibitory concentration (MIC) was defined as the minimum tested concentration that inhibited *C. gloeosporioides* growth for 7 days [13].

With MIC values for thymol and carvacrol, a checkerboard array [35] was used to evaluate the effects of binary mixtures on *C. gloeosporioides* growth. Edible films were prepared following the same methodology, at pH 5. MIC values were transformed into fractional inhibitory concentration (FIC):(1)$${\mathrm{FIC}}_{\mathrm{Thymol}}=\frac{\mathrm{MIC}\;\mathrm{of}\;\mathrm{Thymol}\;\mathrm{in}\;\mathrm{presence}\;\mathrm{of}\;\mathrm{Carvacrol}}{\mathrm{MIC}\;\mathrm{of}\;\mathrm{Thymol}}$$
(2)$${\mathrm{FIC}}_{\mathrm{Carvacrol}}=\frac{\mathrm{MIC}\;\mathrm{of}\;\mathrm{Carvacrol}\;\mathrm{in}\;\mathrm{presence}\;\mathrm{of}\;\mathrm{Thymol}}{\mathrm{MIC}\;\mathrm{of}\;\mathrm{Carvacrol}}$$

Fractional inhibitory concentration index (FIC_Index_) was calculated from FIC values for each antimicrobial as follows:(3)$${\mathrm{FIC}}_{\mathrm{Index}}={\mathrm{FIC}}_{\mathrm{Thymol}}+{\mathrm{FIC}}_{\mathrm{Carvacrol}}$$

Based on the above, an FIC ≤ 0.5 was interpreted as a synergistic effect, 0.5 ≤ FIC ≤ 1 represented as an additive effect, FIC ≤ 4 represented as no interactive effect, and FIC > 4 indicated an antagonistic effect between two tested antimicrobials [20,21].

### 2.4. In Vivo Evaluation of Thymol and Carvacrol Added to Starch Edible Coatings against C. gloeosporioides

Before coating treatments, each fruit was sprayed with a prepared suspension of *C. gloeosporioides.* (with the same spore concentrations that were reported in Section 2.3) and stored at room temperature for 2 h. At this point, fruits were immersed in the coating solution twice for 2 min and dried at room temperature (20 °C). Edible coatings were prepared according to the methodology described previously for film preparation. According to in vitro results, two combinations of thymol and carvacrol were prepared (formulation 1:750 mg/L of carvacrol and 750 mg/L of thymol; Formulation 2:1125 mg/L of carvacrol and 375 mg/L of thymol). Two control were considered in this study (control 1: fruits without coating; control 2: fruits added with an edible coating without antimicrobials). Control and coated samples were stored at 20 °C for 18 days. Physicochemical and fungal analyses were done every 3 days for 18 days. All experiments were performed in triplicate.

Samples firmness (N) was assayed using a handheld hardness tester according to the method reported by Cao et al. [36] from different parts of the fruits in five predetermined positions. For physicochemical characteristics, each sample (approximately 10 g) was blended with 50 mL of distilled water for 1 min using a domestic food blender. The soluble solids concentration (SS) and titratable acidity (TA) were analyzed following the 932.12 and 981.12 methods of the AOAC [37], respectively. Maturity index (MI) was determined as a ratio of total soluble solids and titratable acidity. External injuries were quantified by visual observation on fruit surfaces.

Color parameters of the CIELab scale, *L** (luminosity), *a** (+ red, − green), and *b** (+ yellow, − blue), were measured using a precise colorimeter reader (TCR 200, TIME High Technology, Beijing, China). Color evaluation was conducted in five different points over fruit surfaces. With these parameters, color change (Δ*H*) was calculated following Equation (4):(4)$$\Delta H=\sqrt{{\left(L_{f}^{*}-L_{0}^{*}\right)}^{2}+{\left(a_{f}^{*}-a_{0}^{*}\right)}^{2}+{\left(b_{f}^{*}-b_{0}^{*}\right)}^{2}}$$
where $L_{0}^{*}$, $a_{0}^{*}$, $b_{0}^{*}$ are color parameters of fruits surfaces at the beginning and the end $\left(L_{f}^{*},\;a_{f}^{*},b_{f}^{*}\right)$ of storage [9].

For fungal analyses, 10 g of fruit sample was aseptically removed and directly transferred to 90 mL of peptone water. Decimal dilutions were carried out with peptone water. One mL of each dilution was plated on PDA, and Petri dishes were incubated at 25 °C for 7 days. Colonies were counted, and the results were expressed as log (CFU/g).

#### Data Modeling of *C. gloeosporioides* Growth on In Vivo Evaluation

Growth data of fungal analyses from control and coated samples stored at 20 °C for 18 days were modeled using the modified Gompertz equation [38]:(5)$$Ln\left(\frac{N}{N_{o}}\right)=A*\exp\left\{-\exp\left[\nu_{max}*\exp\left(1\right)/A)\left(\lambda -t\right)+1\right]\right\}$$
where: *N* is the CFU/g at a given time *t* (day), *No* (CFU/g) at the initial time; *A* is the maximum mold growth achieved at the stationary phase, *υ_max_* is the maximum specific growth rate (1/day), and *λ* is the lag phase (day).

### 2.5. Statistical Analysis

To compare the physicochemical and Gompertz parameters, one-way ANOVA was performed by Minitab 15 software (Minitab Inc., State College, PA, USA, 2008). Differences between treatment means were analyzed by Tukey’s comparison test (*p* < 0.05)

## 3. Results and Discussion

### 3.1. In Vitro Evaluation of Thymol and Carvacrol Added to Starch Edible Films against C. gloeosporioides

MIC for *C. gloeosporioides* with carvacrol and thymol added to edible starch films at different pH are shown in Table 1. Both compounds showed similar values except for pH 7 when carvacrol need more than 4000 mg/L. Results demonstrate that edible starch films can serve as carriers to release antimicrobials onto the surface; controlling fungal growth and reducing diffusion into the agar since carvacrol and thymol forms part of the film structure and interacts with the polymer and the plasticizer [39]. In this case, amylose polymer and pH affect the release of carvacrol and thymol due to many factors like electrostatic interactions, osmosis, structural changes, and environmental conditions [19,33]. However, Liolios et al. [40] mentioned that for an adequate operation of edible films added with antimicrobials, the optimal pH should be considered. This condition may maintain a state of dissociation that allows better antimicrobial activity. Martínez-Graciá et al. [41] reported that pH dissociation values of carvacrol and thymol are acidic (4.5 and 5.5, respectively). Acevedo-Fani et al. [42] reported that thymol and carvacrol at proper pH values could change membrane permeability due to molecules binding to proteins by hydrophobic interactions.

With the results obtained, carvacrol and thymol binary mixtures were incorporated into edible films at pH 5. It can be observed that from 9 combinations, 4 show *C. gloeosporioides* growth (Table 2). FIC_index_ values for inhibitory combinations were determined and according to the criteria established by Hossain et al. [21] and Pinto et al. [20]. For binary mixtures, there were three mixes with no interactive effect (750/1125, 1125/750 and 1125/1125 mg/L of carvacrol/thymol, respectively) and two with additive effect (750/750 and 1125/375 mg/L of carvacrol/thymol, respectively).

According to López-Malo et al. [35], binary mixtures from natural antimicrobials are poorly understood. Various compounds with different inactivation mechanisms could achieve adequate microbial control; additive or synergistic are preferred to antagonist effect due to that the effectiveness may be considerably reduced. Phenolic mixtures may increase the number, pores duration by binding these compounds with other proteins or enzymes embedded in the cell membrane. A synergistic effect could be achieved when one of the components probably disintegrates the lipid membrane and makes it easier for the other molecule to enter the cytoplasm. Antimicrobials decrease the cell capacity to tolerate the disturbing effects of the natural antimicrobials on surface characteristics and fungal structure [3,43,44]. Currently, Jahani et al. [11] mentioned that phenolic compounds (like carvacrol and thymol) could affect the normal physiology activity, probably through inhibiting glycolysis, which in turn influence cell energy metabolism, breaking original balance. With additive combinations, edible coatings were developed for the in vivo study.

### 3.2. In Vivo Evaluation of Thymol and Carvacrol Added to Starch Edible Coatings against C. gloeosporioides

Results of physicochemical analyses of mango and papaya treated with edible starch coatings added with carvacrol and thymol are presented in Figure 1 and Figure 2. Firmness (Figure 1A and Figure 2A) represents a key parameter that shows the maturation process in both climacteric fruits. During storage, a significant softening delay (*p* < 0.05) is presented in this parameter in both fruits coated with edible coatings added with carvacrol and thymol. Change in firmness is one of the critical indicators of fruit quality and is closely affected by ripening. Maringgal et al. [1] proposed two possible mechanisms of softening. The first one is due to a breakdown of polymeric carbohydrates that form cell walls; the second is related to water loss causing less cell turgor [45]. Active edible coatings can reduce the oxygen permeability and consequently the respiration rate, which is the inverse related way with the maturity index. Thus, an increase in the respiration rate and increase of maturity index is observed (Figure 1B and Figure 2B). During ripening, fruit acid content decreases due to their use in some metabolic pathways and increases total soluble solids [46,47]. In uncoated mango and papaya and starch coated fruits presented a significative (*p* > 0.05) maturity index that represents higher metabolic respiration. Starch coating posses a suitable water vapor resistance, which decreases the degree of dehydration. Sahraee et al. [48] mentioned that this characteristic might be due to their considerable number of hydrogen bonds along the film, which helps adjacent chains bind tightly to each other. This effect prevents ethylene production and a decrease in O_2_ and CO_2_ permeability. Moreover, at the end of the storage period, the color change is significative higher (*p* < 0.05) in uncoated fruits than coated ones (Figure 1D and Figure 2D). Although edible starch coatings did not change the fruit’s surface color (because of their transparent characteristic), *L** values increased. For mango *L** *a** *b** values change for green to yellow during storage (*L** 62.99 ± 3.32 to 83.03 ± 4.32; *a** −4.84 ± 1.03 to 10.07 ± 1.34; *b** 6.85 ± 1.67 to 80.26 ± 3.45). In papaya from green to orange (*L** 52.69 ± 4.57 to 64.94 ± 2.46; *a** −20.21 ± 2.25 to 21.48 ± 2.11; *b** 49.65 ± 4.23 to 56.01 ± 5.27). Md Nor et al. [30] specified that mango exhibits the tendency to browning disorder after harvest. High oxygen concentration can stimulate polyphenol oxidase (PPO) activity and produce a rapid browning reaction. Coated fruit reduces the available oxygen concentration retard PPO activity, and regulate the undesirable peel fruit browning. Some compounds, like thymol and carvacrol, can improve PPO inhibition. Moreover, climacteric fruits peel color changes due to a transformation of carotenoids from chloroplasts to chromoplasts [49]. Furthermore, Maringgal et al. [1] mentioned that edible coatings improve fruits’ appearance by maintaining the color and glossy surface. Our results show a moderate color change development in coated fruits during storage. This effect could be attributed to the lower respiration rate; caused by a delay in ethylene production by the modification of the fruit atmosphere. Yousuf et al. [50] quoted that color values viz *L**, *a**, *b** and subsequently, the parameters derived (chroma and hue angle). Changes will occur upon storage, but the variation will depend upon the particular fruit into consideration. Edible coatings containing mixtures of carvacrol and thymol are suitable for delaying ripening because these natural compounds are continuously released over time on the fruit surface, delaying firmness and peel color changes on papaya and mango [51,52]. Finally, an indirect parameter that confirms fungal growth is the number of lesions detected on the surface of the fruit (Figure 1C and Figure 2C).

The effect of carvacrol and thymol binary mixtures added on edible starch coatings on *C. gloeosporioides* growth on mango and papaya surfaces is shown in Figure 3. Although in vitro assays edible films demonstrated an antifungal effect, when they were applied as coatings, they maintained a fungistatic effect on *C. gloeosporioides* growth. Guimarães et al. [4] pointed out that fruit (species and even cultivar) and the environmental conditions may explain why results from in vivo experiments cannot often be anticipated by the in vitro activity. Similar findings were also reported by Pinto et al. [20] that found the different antifungal activity of thyme essential oil between in vitro and in vivo experiments.

It can be observed with Gompertz parameters (Table 3) that, for both fruits, Formulation 2 is the one with a significant and high fungistatic effect (*p* < 0.05). This formulation increased the lag phase and reduced th maximum specific growth rate in both fruits. These differences between both analysis orders are due to time and storage conditions that allow a gradual loss of carvacrol and thymol, making it only a delay on fruit surface fungal growth. Nevertheless, the shelf life of both fruits stored at room temperature can be extended for at least double the time as those that are uncoated. Similar results were obtained with chitosan coatings added with mint essential oil on mango var. Atkins [51], and with a combination of diverse mint varieties, essential oils in papaya [10].

## 4. Conclusions

Carvacrol and thymol incorporated into edible films and coatings are able to reduce the incidence of anthracnose symptoms on mango and papaya. Moreover, active coatings preserve fruit quality as long as possible by reducing senescence rate, acting quite similar to a modified atmosphere packaging. The main effect in vitro is fungicidal, while in vivo is fungistatic. Both antimicrobials have an additive effect on *C. gloeosporioides* development. The use of natural antimicrobials incorporated into edible films and coatings could represent an alternative to reduce postharvest losses in important tropical fruits.

## Figures and Tables

**Figure 1 foods-10-00175-f001:**
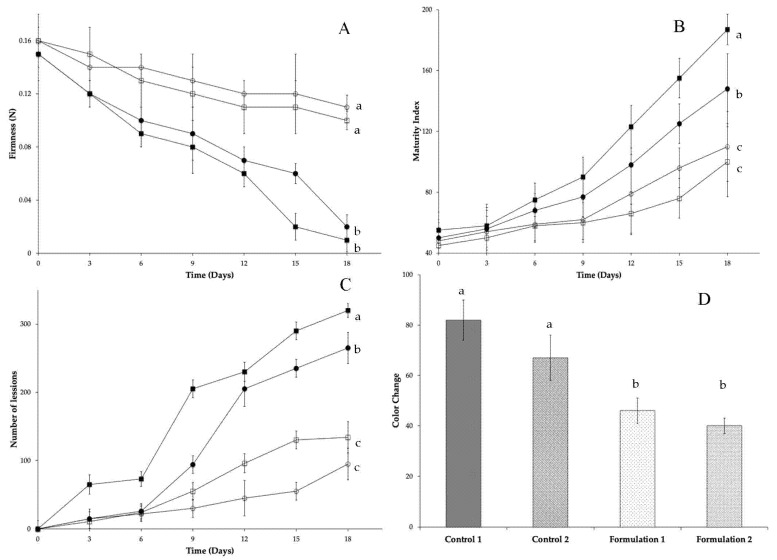
Effect of carvacrol and thymol binary mixtures added on edible starch coatings on mango (control 1 (■), control 2 (●), formulation 1:750 mg/L of carvacrol and 750 mg/L of thymol (□), formulation 2:1125 mg/L of carvacrol and 375 mg/L of thymol (O)) firmness (**A**), maturity index (**B**), surface lesions (**C**) and color change (**D**). Different superscript letters within for each treatment are significantly different (*p* < 0.05).

**Figure 2 foods-10-00175-f002:**
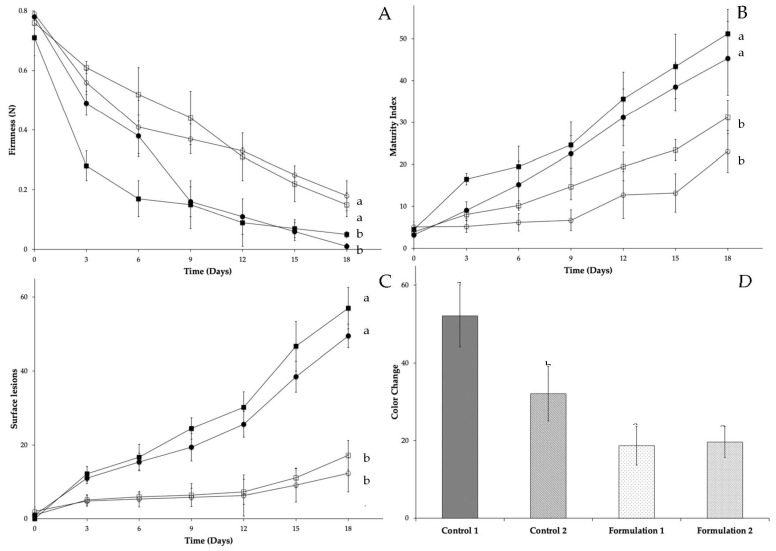
Effect of carvacrol and thymol binary mixtures added on edible starch coatings on papaya (control 1 (■), control 2 (●), formulation 1: 750 mg/L of carvacrol and 750 mg/L of thymol (□), formulation 2:1125 mg/L of carvacrol and 375 mg/L of thymol (O)) firmness (**A**), maturity index (**B**), surface lesions (**C**) and color change (**D**). Different superscript letters within for each treatment are significantly different (*p* < 0.05).

**Figure 3 foods-10-00175-f003:**
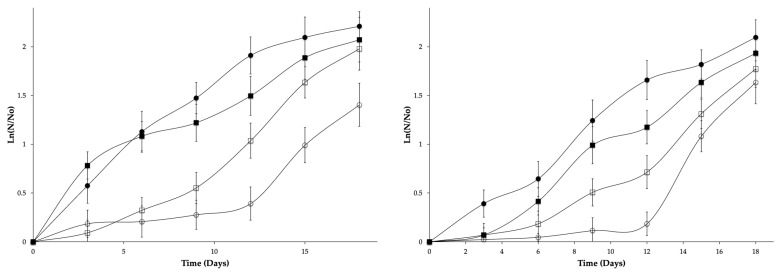
Effect of carvacrol and thymol binary mixtures added on edible starch coatings on *C. gloeosporioides* growth on mango (**A**) and papaya (**B**) surfaces (control 1 (■), control 2 (●), formulation 1:750 mg/L of carvacrol and 750 mg/L of thymol (□), formulation 2:1125 mg/L of carvacrol and 375 mg/L of thymol (O)).

**Table 1 foods-10-00175-t001:** Minimum inhibitory concentration (MIC) of *C. gloeosporioides* at selected concentrations of carvacrol and thymol added to edible starch films at different pH values.

pH	Carvacrol (mg/L)	Thymol (mg/L)
4	1750	1750
5	1500	1500
6	2000	3000
7	3000	>4000

**Table 2 foods-10-00175-t002:** *C. gloeosporioides* growth (G) or no growth (NG) response at combinations of carvacrol and thymol (mg/L) added to edible starch films at pH 5. NG response with FIC_index_ in parenthesis.

	Thymol (mg/L)		
Carvacrol (mg/L)	375	750	1125
375	G	G	G
750	G	NG (0.85)	NG (1.07)
1125	NG (0.85)	NG (1.07)	NG (1.28)

**Table 3 foods-10-00175-t003:** Modified Gompertz model parameters (mean ± standard deviation) for *C. gloeosporioides* growth on mango and papaya surfaces with carvacrol and thymol binary mixtures added to edible starch coatings.

Mango				Papaya			
	A	υ_max_ (1/day)	λ (day)		A	υ_max_ (1/day)	λ (day)
Control 1	2.14 ± 0.23 ^a^	0.27 ± 0.05 ^a^	0.59 ± 0.06 ^a^	Control 1	1.97 ± 0.23 ^a^	0.37 ± 0.08 ^a^	0.76 ± 0.17 ^a^
Control 2	2.05 ± 0.11 ^a^	0.33 ± 0.07 ^a^	0.76 ± 0.12 ^a^	Control 2	1.88 ± 0.16 ^a^	0.42 ± 0.03 ^a^	3.11 ± 0.12 ^b^
Formulation 1	1.92 ± 0.14 ^a^	0.22 ± 0.03 ^b^	4.11 ± 0.62 ^c^	Formulation 1	1.72 ± 0.14 ^a,b^	0.24 ± 0.05 ^b^	5.68 ± 0.56 ^c^
Formulation 2	1.46 ± 0.18 ^b^	0.17 ± 0.06 ^b^	8.96 ± 0.89 ^d^	Formulation 2	1.61 ± 0.08 ^b^	0.21 ± 0.07 ^b^	13.43 ± 1.41 ^d^

A: maximum growth in the stationary phase; υ_max_: maximum specific growth rate; λ: lag phase. Means followed by a different superscript letter within a column for each are significantly different (*p* < 0.05).

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
