# Peer review of "Starch Edible Films/Coatings Added with Carvacrol and Thymol: In Vitro and In Vivo Evaluation against Colletotrichum gloeosporioides"

_foods, 2021, doi:10.3390/foods10010175_

Round 1
Reviewer 1 Report
Manuscript: 1056926
Journal: Foods
Title: Starch edible films/coatings added with carvacrol and thymol: In vitro and in vivo evaluation against Colletotrichum gloeosporioides
The paper describes the effect of starch films added with carvacrol and thymol combinations against Colletotrichum gloesporioides in mango and papaya fruit. Fruit lesions, colour change, firmness, maturity index and growth parameters were evaluated. The manuscript is quite interesting. Several papers deal with the use of antimicrobial edible coatings to control anthracnose on banana, mango, and papaya. However, few reports are available using different combinations of essential oils. I suggest a minor revision for a better presentation of the results.
Minor comments
L18-19 Minimal Inhibitory Concentration (MIC) of carvacrol and thymol was determined at different pH values against Colletotrichum gloesporioides
L24 According to recent works a FICI value of 0.85 is not sufficient to indicate synergism. Please follow the works of Hossain et al. 2016 (X) and Pinto et al. 2020 (Y) to determine the interactions. In this case the effect is additive. Please correct throughout the manuscript.
X Hossain, F.; Follett, P.; Vu, K. D.; Harich, M.; Salmieri, S.; Lacroix, M. Evidence for synergistic activity of plant-derived essential oils against fungal pathogens of food. Food microbiology 2016, 53, 24-30.
L28-29 These results were corroborated by the increase of the lag phase and the reduction of the growth rate
L36 are recognized as antimicrobials
L36-37 Please remove
L39 Please add the following sentence:
For instance, thymol showed antifungal activity in vapor phase (A), with synergistic effects with other terpenes against different fungal genera (B).
A Pinto L.; Cefola M.; Bonifacio M. A.; Cometa S.; Bocchino C.; Pace B.; De Giglio E.; Palumbo M.; Sada A.; Logrieco A.F.; Baruzzi F. Effect of red thyme oil (Thymus vulgaris L.) vapours on fungal decay, quality parameters and shelf life of oranges during cold storage. Food Chemistry 2021, 336C, 127590.
L39 and L43 Delete semi column
L49 Other papers could be mentioned. The following are suggested:
Batista, D. D. V. S., Reis, R. C., Almeida, J. M., Rezende, B., Bragança, C. A. D., & da Silva, F. (2020). Edible coatings in post-harvest papaya: impact on physical–chemical and sensory characteristics. Journal of Food Science and Technology, 57(1), 274-281.
Tavassoli-Kafrani, E., Gamage, M. V., Dumée, L. F., Kong, L., & Zhao, S. (2020). Edible films and coatings for shelf life extension of mango: a review. Critical Reviews in Food Science and Nutrition, 1-29.
L50-L65 Please move this part at the begin of the introduction
L65-67 Please expand the aim of the work with other details (i.e., binary combinations, additive effects and so on).
L71 were purchased…
L99 Delete semi column
L102 when the solution reached 40°C…
L122-123 Please add the interpretation of the results of the FIC index (FICI value) according to Hossain et al. 2016 and Pinto et al. 2020.
L131 Two controls were considered in this study…
L131-132 Control and coated samples were stored at 20°C….
L152-157 Please add details in this section. Which data were used for the modelling? Which conditions? The collection of the growth data should be detailed
L159 analysis of variance. ANOVA? One or two-way ANOVA? Please add details
L172 Delete semi column
L172-173 Please divide in two sentences
L182-183 Please delete Davidson and Parish and replace with Hossain et al. 2016 and Pinto et al. 2020
L182-185 Please revise according to the new references
L193-196 Please divide in two sentences
L196-199 Please rewrite this sentence. It is not clear
L200 Please revise the sentence according to the results
L211 Delete respiration rate. The figure reports only maturity index
L213-214 In uncoated papaya and mango….
L213-215 Please highlights that formulation 2 showed a reduction of the maturity index in comparison to other samples.
L217 Replace dot with comma. Delete were…
L218-223 Please rewrite these sentences. They are not clear
L224 Delete semi column
L263 Please add the following sentence:
Similar findings were also reported by Pinto et al. (A) that found different antifungal activity of thyme essential oil between in vitro and in vivo experiments.
A Pinto L.; Cefola M.; Bonifacio M. A.; Cometa S.; Bocchino C.; Pace B.; De Giglio E.; Palumbo M.; Sada A.; Logrieco A.F.; Baruzzi F. Effect of red thyme oil (Thymus vulgaris L.) vapours on fungal decay, quality parameters and shelf life of oranges during cold storage. Food Chemistry 2021, 336C, 127590.
L274-275 This formulation increased the lag phase and reduced the maximum specific growth rate in both fruit.
L289 Revise the conclusion
L290 the use of antimicrobials included into edible films…
L290 Delete further studies
Please add the statistical analysis in Fig. 1, 2, and 3 indicating significant differences with symbols (*) or letters (a, b, c…).
Author Response
REVIEWER 1
Journal: Foods
Title: Starch edible films/coatings added with carvacrol and thymol: In vitro and in vivo evaluation against Colletotrichum gloeosporioides
The paper describes the effect of starch films added with carvacrol and thymol combinations against Colletotrichum gloesporioides in mango and papaya fruit. Fruit lesions, colour change, firmness, maturity index and growth parameters were evaluated. The manuscript is quite interesting. Several papers deal with the use of antimicrobial edible coatings to control anthracnose on banana, mango, and papaya. However, few reports are available using different combinations of essential oils. I suggest a minor revision for a better presentation of the results.
Minor comments
L18-19 Minimal Inhibitory Concentration (MIC) of carvacrol and thymol was determined at different pH values against Colletotrichum gloesporioides
ANSWER: Corrected L19-20
L24 According to recent works a FICI value of 0.85 is not sufficient to indicate synergism. Please follow the works of Hossain et al. 2016 (X) and Pinto et al. 2020 (Y) to determine the interactions. In this case the effect is additive. Please correct throughout the manuscript.
X Hossain, F.; Follett, P.; Vu, K. D.; Harich, M.; Salmieri, S.; Lacroix, M. Evidence for synergistic activity of plant-derived essential oils against fungal pathogens of food. Food microbiology 2016, 53, 24-30.
ANSWER: Corrected, we appreciated the valuable information, and the term were corrected throughout the manuscript.
L28-29 These results were corroborated by the increase of the lag phase and the reduction of the growth rate
ANSWER: Corrected
L36 are recognized as antimicrobials
ANSWER: Corrected L52
L36-37 Please remove
ANSWER: Line removed
L39 Please add the following sentence:
For instance, thymol showed antifungal activity in vapor phase (A), with synergistic effects with other terpenes against different fungal genera (B).
A Pinto L.; Cefola M.; Bonifacio M. A.; Cometa S.; Bocchino C.; Pace B.; De Giglio E.; Palumbo M.; Sada A.; Logrieco A.F.; Baruzzi F. Effect of red thyme oil (Thymus vulgaris L.) vapours on fungal decay, quality parameters and shelf life of oranges during cold storage. Food Chemistry 2021, 336C, 127590.
ANSWER: Sentence added with the references L61-62
L39 and L43 Delete semi column
ANSWER: Semi columns deleted
L49 Other papers could be mentioned. The following are suggested:
Batista, D. D. V. S., Reis, R. C., Almeida, J. M., Rezende, B., Bragança, C. A. D., & da Silva, F. (2020). Edible coatings in post-harvest papaya: impact on physical–chemical and sensory characteristics. Journal of Food Science and Technology, 57(1), 274-281.
Tavassoli-Kafrani, E., Gamage, M. V., Dumée, L. F., Kong, L., & Zhao, S. (2020). Edible films and coatings for shelf life extension of mango: a review. Critical Reviews in Food Science and Nutrition, 1-29.
ANSWER: Papers suggested by the reviewer were mentioned
L50-L65 Please move this part at the begin of the introduction
ANSWER: Sentences were moved to the beginning L34-49
L65-67 Please expand the aim of the work with other details (i.e., binary combinations, additive effects and so on).
ANSWER: Corrected L79-81
L71 were purchased…
ANSWER: Corrected L84-85
L99 Delete semi column
ANSWER: Semi column deleted L112
L102 when the solution reached 40°C…
ANSWER: Corrected line 115
L122-123 Please add the interpretation of the results of the FIC index (FICI value) according to Hossain et al. 2016 and Pinto et al. 2020.
ANSWER: Interpretation added L136-138
L131 Two controls were considered in this study…
ANSWER: Corrected L147
L131-132 Control and coated samples were stored at 20°C….
ANSWER: Corrected L149
L152-157 Please add details in this section. Which data were used for the modelling? Which conditions? The collection of the growth data should be detailed
ANSWER: Corrected L170-171
L159 analysis of variance. ANOVA? One or two-way ANOVA? Please add details
ANSWER: Corrected L177
L172 Delete semi column
ANSWER: Semi column deleted
L172-173 Please divide in two sentences
ANSWER: Sentence was divided L189-191
L182-183 Please delete Davidson and Parish and replace with Hossain et al. 2016 and Pinto et al. 2020
ANSWER: Corrected L199-202.
L182-185 Please revise according to the new references
ANSWER: Paragraph corrected L199-202
L193-196 Please divide in two sentences:
ANSWER: Corrected L210-213
L196-199 Please rewrite this sentence. It is not clear
ANSWER: Sentence rewrite L213-215
L200 Please revise the sentence according to the results
ANSWER: Corrected L216
L211 Delete respiration rate. The figure reports only maturity index
ANSWER: Corrected L227
L213-214 In uncoated papaya and mango….
ANSWER: Corrected L 229
L213-215 Please highlights that formulation 2 showed a reduction of the maturity index in comparison to other samples.
ANSWER: Corrected L231-234
L217 Replace dot with comma. Delete were…
ANSWER: Corrected
L218-223 Please rewrite these sentences. They are not clear
ANSWER: Corrected L244-247
L224 Delete semi column
ANSWER: Corrected
L263 Please add the following sentence:
Similar findings were also reported by Pinto et al. (A) that found different antifungal activity of thyme essential oil between in vitro and in vivo experiments.
A Pinto L.; Cefola M.; Bonifacio M. A.; Cometa S.; Bocchino C.; Pace B.; De Giglio E.; Palumbo M.; Sada A.; Logrieco A.F.; Baruzzi F. Effect of red thyme oil (Thymus vulgaris L.) vapours on fungal decay, quality parameters and shelf life of oranges during cold storage. Food Chemistry 2021, 336C, 127590.
ANSWER: Sentence and reference added L261-263
L274-275 This formulation increased the lag phase and reduced the maximum specific growth rate in both fruit.
ANSWER: Corrected L290-291
L289 Revise the conclusion
ANSWER: Conclusion rewrite L303-309
L290 the use of antimicrobials included into edible films…
ANSWER: Corrected
L290 Delete further studies
ANSWER: Corrected
Please add the statistical analysis in Fig. 1, 2, and 3 indicating significant differences with symbols (*) or letters (a, b, c…).
ANSWER: Statistical analysis were added on Figures 1 and 2. Statistical analysis of Figure 3 is reported on Gompertz parameters in Table 3.
Reviewer 2 Report
Overall, the manuscript is interesting and brings interesting new insights into the application of new edible food protection packaging. However, there is a significant calculation error in the methodology that must be corrected.
- Lines 153-157: The modified Gompertz equation (line 152) used is invalid. In Char, C .; Guerrero, S .; Alzamora, S. Growth of Eurotium chevalieri in milk jam: influence of pH, potassium sorbate and water activity. J. Food Saf. 2007, 27 is the natural logarithm (Ln), not the decimal (log). The formula was also presented in the publication: Dalgaard and Koutsoumanis (2001): Comparison of maximum specific growth rates and lag times estimated from absorbance and viable count data by different mathematical models. J. Microbiol. Methods 43 (2001) 183–196.
- There is a fundamental difference between “Ln” and “Log10” and therefore there is a calculation error.
- Lines 273-285 with Table 3: The results discussed and presented in Table 3 should be recalculated.
Author Response
REVIEWER 2
Overall, the manuscript is interesting and brings interesting new insights into the application of new edible food protection packaging. However, there is a significant calculation error in the methodology that must be corrected.
- Lines 153-157: The modified Gompertz equation (line 152) used is invalid. In Char, C .; Guerrero, S .; Alzamora, S. Growth of Eurotium chevalieri in milk jam: influence of pH, potassium sorbate and water activity. J. Food Saf. 2007, 27 is the natural logarithm (Ln), not the decimal (log). The formula was also presented in the publication: Dalgaard and Koutsoumanis (2001): Comparison of maximum specific growth rates and lag times estimated from absorbance and viable count data by different mathematical models. J. Microbiol. Methods 43 (2001) 183–196.
- There is a fundamental difference between “Ln” and “Log10” and therefore there is a calculation error.
- Lines 273-285 with Table 3: The results discussed and presented in Table 3 should be recalculated.
ANSWER: Equation was corrected according to Ln instead of Log10. Therefore, values were recalculated and modified Figure 3 and Table 3.
Reviewer 3 Report
Introduction section
As the manuscripts main propose is the use of starch biopolymer as coating for fruit, the introduction section should include at least a paragraph about the interest of using starch as coating, why starch and not other biopolymer (i.e.: chitosan, caseinates, etc.).
Since glycerol has been used as plasticizer the state of the art should be mentioned, why it was selected? is it biobased), etc.
Since there are a lot of manuscript reporting edible polymers with carvacrol and thymol, the state of the art in this field should be summarized highlighting the main differences proposed here
Materials and methods
Line 99. why high amylose corn starch was selected in this study. Please clarify this point in the introduction section of the mansucript.
Line 106 the first time that FFS is mentioned should be explained.
Results
- The colorimetric results should be extended and better discussed. Some comments regarding the a* (red-green) and b* (yellow-blue) coordinates results should be discussed.
- as the color changes in fruits is very important for consumer acceptance. Photographs of fruits with and without the coatings should be added. Moreover, the browning effect should be considered since fresh fruit browning is caused by enzymatic oxidation of phenolic compounds mediated by polyphenol oxidase activity, and two strategies for inhibiting this process are through the reduction of oxygen and addition of antioxidants. Thus, considering that thymol and carvacrol based coating can potentially show antioxidant performance it is highly recommended to add some comments of the antibrowning effect produced by these coatings.
- The statistic analysis of results reported in Figure 1, 2 and 3 should (have to) be added.
- The quality of Figure 1, 2 and 3 should be improved.
References
The references should be updated considering the state of the art of the main topic of the present research (edible polymeric films with essential oils).
Author Response
REVIEWER 3
Introduction section
As the manuscripts main propose is the use of starch biopolymer as coating for fruit, the introduction section should include at least a paragraph about the interest of using starch as coating, why starch and not other biopolymer (i.e.: chitosan, caseinates, etc.).
ANSWER: Information was added L72-78
Since glycerol has been used as plasticizer the state of the art should be mentioned, why it was selected? is it biobased), etc.
ANSWER: Information was added L76-78
Since there are a lot of manuscript reporting edible polymers with carvacrol and thymol, the state of the art in this field should be summarized highlighting the main differences proposed here
ANSWER: Information was added L53-59.
Materials and methods
Line 99. why high amylose corn starch was selected in this study. Please clarify this point in the introduction section of the mansucript.
ANSWER: Information was clarified in L72-74
Line 106 the first time that FFS is mentioned should be explained.
ANSWER: An explanation of FFS abbreviature was added L113-114
Results
- The colorimetric results should be extended and better discussed. Some comments regarding the a* (red-green) and b* (yellow-blue) coordinates results should be discussed.
ANSWER: Discussion of colorimetric results were added. L237-251
- as the color changes in fruits is very important for consumer acceptance. Photographs of fruits with and without the coatings should be added. Moreover, the browning effect should be considered since fresh fruit browning is caused by enzymatic oxidation of phenolic compounds mediated by polyphenol oxidase activity, and two strategies for inhibiting this process are through the reduction of oxygen and addition of antioxidants. Thus, considering that thymol and carvacrol based coating can potentially show antioxidant performance it is highly recommended to add some comments of the antibrowning effect produced by these coatings.
ANSWER: Browning effect is discussed L240-251. Unfortunately photographs are not complete because my student were robed, and the ones we have are with a very low quality.
- The statistical analysis of results reported in Figure 1, 2 and 3 should (have to) be added.
ANSWER: Statistical analysis of results were added on Figures 1 and 2. Statistical analysis of Figure 3 is reported on Gompertz parameters in Table 3.
- The quality of Figure 1, 2 and 3 should be improved.
ANSWER: Figures quality was improved
References
The references should be updated considering the state of the art of the main topic of the present research (edible polymeric films with essential oils).
ANSWER: References were updated.
Round 2
Reviewer 1 Report
The authors revised the manuscript according to Reviewer's comments.
Author Response
The authors revised the manuscript according to Reviewer's comments
Reviewer 2 Report
Line 173: Improve No units (CFU / g)
Line 235: if the differences were significant statistically then it should be (p <0.05)
Author Response
Line 173: Improve No units (CFU / g)
ANSWER: Corrected
Line 235: if the differences were significant statistically then it should be (p <0.05)
ANSERR: Corrected